# Selection and Evaluation of *Porcine circovirus* (PCV) 2d Vaccine Strains to Protect against Currently Prevalent PCV2

**DOI:** 10.3390/vaccines11091447

**Published:** 2023-09-01

**Authors:** Lanjeong Ju, Su-Hwa You, Min-A Lee, Usharani Jayaramaiah, Young-Ju Jeong, Hyang-Sim Lee, Bang-Hun Hyun, Nakhyung Lee, Seok-Jin Kang

**Affiliations:** 1Division of Viral Diseases, Animal and Plant Quarantine Agency, 177 Hyeoksin 8-ro, Gimcheon-si 39660, Gyeongsangbuk-do, Republic of Korea; lanjeong@korea.kr (L.J.); ysh0108@korea.kr (S.-H.Y.); ma5147@korea.kr (M.-A.L.); ushavet85@gmail.com (U.J.); leehs76@korea.kr (H.-S.L.); hyunbh@korea.kr (B.-H.H.); 2Technology Institute, KBNP, Anyang-si 14059, Gyeonggi-do, Republic of Korea; yjjeong@kbnp.co.kr (Y.-J.J.); nhlee21@kbnp.co.kr (N.L.)

**Keywords:** PCV2, VLP, vaccine, cross-neutralization, in vivo protection

## Abstract

*Porcine circovirus* (PCV) 2d is a common genotype in South Korea, and the cross-protective ability of PCV2a-based vaccines has been reported recently. In this study, a PCV2d vaccine candidate was selected, and its protective efficacy against the PCV2d isolate was evaluated. From 2016 to 2020, 234 PCV2d isolates were phylogenetically analyzed using open reading frame 2 (ORF2) sequences and classified into four subgroups: PCV2d-1, PCV2d-2, PCV2d-3, and PCV2d-4. Except for PCV2d-4, which consisted of ungrouped isolates, the three subgroups showed distinct differences at amino acid positions 53 and 169 in the ORF2. The detection rates of PCV2d-1, PCV2d-2, and PCV2d-3 were 36.5, 37.4, and 3.7%, respectively, and representative isolates were selected from each subgroup (QIA244, QIA126, and QIA169, respectively). In the neutralization assay, QIA244 showed the lowest neutralization efficiency among the three PCV2a-based vaccines, whereas the virus-like particles of QIA244 (rQIA244) provided broader protection against the three genotypes than did those of QIA126 and rQIA169. To further evaluate rQIA244 in pigs, the experimental groups were divided into rQIA244-vaccine (2dVac), commercial PCV2a-vaccine (2aVac), and no-vaccination (noVac) groups. The 2dVac effectively reduced the copy number of PCV2d in blood and tissues, as well as in tissue lesions, compared to the effect of 2aVac. Collectively, 2dVac provided by QIA244 ORF2 successfully demonstrated protective efficacy against the currently prevalent PCV2d in vitro neutralization and in vivo assays.

## 1. Introduction

*Porcine circovirus* (PCV) was identified in 1982 as a cell culture contaminant of porcine kidney cells [1,2]. PCV belongs to the family *Circoviridae*, which includes PCV1, PCV2, PCV3, and PCV4. It is a single-stranded circular DNA virus with an icosahedral symmetric capsid (cap). Among the various types of Porcine circoviruses, PCV2 is the most common and important etiological agent of porcine circovirus-associated disease (PCVAD) and is responsible for significant economic losses [3]. Over the last decade, its prevalence has increased, posing a major threat to the swine industry [4].

PCV2 has different genotypes, from PCV2a to PCV2h, based on the *orf2* sequence [5,6,7]. The genotype shifts are a notable phenomenon among the different genotypes of PCV2. A genotype shift from PCV2a to PCV2b and, more recently, to PCV2d, has occurred in major pork-producing countries, including South Korea [8,9,10,11]. In a recent study on PCV2 prevalence in South Korea between 2016 and 2020, PCV2d accounted for approximately 90% of cases [12]. This shift between PCV2 genotypes has led to outbreaks and vaccine failures in the global swine industry [13,14].

The PCV2 cap protein, encoded by *orf2*, is a major immunogenic protein that induces protective immunity against PCV2 infection and plays an important role in binding to the heparan sulfate receptor [15]. The *orf2* gene of PCV2a has a high degree of similarity to that of PCV2b but differs significantly from that of PCV2d, which has spread rapidly throughout the world. Owing to its immunogenicity, this cap protein is a target for PCV2 vaccine development [16].

PCV2a is the main component of most commercially available vaccines used to control PCV2 infections that cause PCVAD [17]. However, the currently circulating genotypes differ from the vaccine genotypes. Although PCV2a-based commercial vaccines provide protection against viral challenges, they fail to reduce the viral load in the blood and tissues. The immunogenic antigens of many commercial PCV vaccines are recombinant proteins containing the PCV2a cap protein produced in insect cells using a baculovirus expression system. Although the manufacturer suggested that PCV2a-based vaccines provide cross-protection against PCV2d, the prevalence of PCV2d in the field cannot be prevented using PCV2a-based vaccines. Previous studies have shown that commercial PCV2a-based vaccines neutralize PCV2b isolates better than PCV2d isolates do [18]. Experimental evidence from controlled trials also supports the idea that homologous vaccines are more effective than heterologous ones at reducing viremia in concurrent PCV2a and PCV2d challenges [19]. Therefore, vaccines with circulating genotypes are needed.

The shift in PCV2 genotypes and vaccine failures necessitates the development of a vaccine specific to the currently circulating genotype and the comparison of its protective efficacy with commercially available vaccines commonly used in Korea. In a previous study, we produced a virus-like particle (VLP) vaccine of the PCV2d ORF2 protein, which demonstrated cross-protection against different genotypes [20]. Designing a vaccine tailored to a specific genotype circulating in a country can help reduce vaccine costs and effectively target prevalent strains. To date, no study has compared recombinant PCV2d-VLP with commercially available PCV2a-based vaccines. In this study, we investigated the ability of a PCV2d vaccine to provide broad protection against PCV2 infection and evaluated the efficacy of the PCV2d vaccine against PCV2d challenge in vivo.

## 2. Materials and Methods

### 2.1. Viruses and Cells

PCV2 viruses were obtained from clinical samples in our previous study [12]. To analyze the genetic relation between the PCV2d isolates detected from pig farms in Korea, we performed multiple sequence alignments using the CLC Main Workbench (Qiagen, Version 7.0.3). The PCV2 strains, QIA215 (PCV2a), QIA418 (PCV2b), and four PCV2d (QIA244, QIA126, QIA169, and QIA416), were propagated and titrated in PCV-free PK15 cells as previously described [20].

### 2.2. Production of Recombinant VLP

A baculovirus expression system (BES) was used for producing ORF2-based VLPs as previously described [20]. PCV2 VLPs designated as rQIA244, rQIA126, and rQIA169 were derived from full-length *ORF2* genes from the QIA244, QIA126, and QIA169 strains, respectively.

### 2.3. Guinea Pig Immunization

For the cross-neutralization test, antisera were prepared by injecting rQIA244, rQIA216, and rQIA169 into guinea pigs (three heads per group). For the first injection, each VLP (200 µg/mL) was mixed with a 10% carbomer (Lubrizol, OH, USA) and injected intramuscularly into the guinea pigs. Two weeks later, a second injection was administered in the same manner as the first. Two weeks after the second injection, all guinea pigs were euthanized, and the sera were collected. The sera were aliquoted and stored at −20 °C until use. The animal experimental protocol for this study was approved by the KBNP Institutional Animal Care and Use Committee (approval number; KBNP 20-5), and all experiments were conducted according to the guidelines suggested by the Animal Ethics Committee.

### 2.4. Pig Studies

Eighteen piglets, which were commercially crossbred, 3 weeks old, antigen-free of *Mycoplasma,* PRRSV and PCV2, and either negative (cut-off < 0.3) or with false positive (cut-off < 0.4) of PCV2-specific antibodies, were allocated into three groups (six pigs per group) considering body weight and ELISA titers and vaccinated the next day (Figure 1). Two groups were vaccinated with 2dVac (rQIA244 + 10% carbomer) and 2aVac (the commercial PCV2a genotype-based vaccine), while the other group served as the noVac group (not vaccinated). All groups were co-challenged with PCV2d (QIA244 genotype) and PRRSV-1 (European type) at 6 weeks of age and euthanized at 21 days after the challenge (dpc), as described previously [20]. The animal study was approved by the KBNP Institutional Animal Care and Use Committee (approval number; KBNP 20-6).

### 2.5. Clinical Signs and Body Weight

After the challenge, all pigs were checked daily for clinical signs according to a routine management program. The body weights were recorded weekly until 21 dpc, and the average daily weight gain (ADWG, kg/d) was calculated from the body weights measured between 0 and 21 dpc.

### 2.6. Quantification of the Virus in Serum and Tissues

For quantifying the genomic copy of PCV2, serum samples were collected from all piglets of −21, −7, 0, 4, 7, 14 and 21 dpc, and tissues such as lung, tonsil, mesenteric lymph node (LN) and inguinal LN were sampled at 21 dpc. Total nucleic acid extraction and quantification of the genomic copy number were described in our previous study [20].

### 2.7. Viral Neutralization Assay

For viral neutralization assay, the sera of guinea pigs and pigs were used. Vaccines A, B and C are commercially available vaccines based on the PCV2a genotype. Sera were collected 6 weeks after immunization of 3-week-old piglets at the indicated doses from each vaccine. Viral neutralization was conducted by a fluorescent focus neutralization (FFN) assay according to the method described by Meerts et al. [21], and our modified virus neuralization test (VNT90) was described in a previous study [12,20].

### 2.8. Gross Pathology and Histopathology

The macro- and microscopic lesions were scored as previously described [20].

### 2.9. Statistical Analyses

The results are presented as the mean ± standard error (SE) based on triplicate experiments. Statistical significance was assessed using the GraphPad Prism 7 (GraphPad Software, Inc., San Diego, CA, USA), utilizing one-way analysis of variance (ANOVA) and Dunnett’s multiple comparison test to evaluated group differences. Statistical significance was set at *p*-value < 0.05.

## 3. Results

### 3.1. Antigenicity of PCV2d Field Isolates in South Korea

#### 3.1.1. Analysis of Cap Proteins in the Recently Prevalent PCV2d

Recently, PCV2d has become increasingly prevalent in South Korea. From 2016 to 2020, 234 PCV2d field strains were phylogenetically analyzed using the *orf2* gene sequence. The analysis revealed that PCV2d isolates could be classified into four subgroups: PCV2d-1, PCV2d-2, PCV2d-3, and PCV2d-4 (Figure 2a and Appendix A). In the amino acid sequence alignment of ORF2, the PCV2d strains showed differences of approximately 10% (similarity: 89.4–89.7%) and 7% (similarity: 92.8–93.1%) compared with those of PCV2a and PCV2b, respectively. However, the inter-PCV2d isolates showed a less than 1% difference (similarity: 99.3–99.9%). Despite high conservation, the ORF2 sequences of the PCV2d isolates showed distinct differences at positions 53 (F or I) and 169 (R or G) (Figure 2b). Among the 234 PCV2d isolates, the detection rate of PCV2d-1 and PCV2d-2 was approximately 73.9% (36.5% and 37.4%, respectively) (Figure 2c). In contrast, PCV2d-3 had low detection rates of 8.6% in 2017 and 7.0% in 2018 (3.7%). PCV2d-4, which was included in the three groups, had an average of 22.4% and showed random mutations throughout the *orf2* gene sequence without grouping based on the difference between positions 53 and 169.

#### 3.1.2. Neutralization of Commercial PCV2a-Based Vaccines against PCV2 Strains

Almost all the currently available commercial vaccines are PCV2a-based. In this study, the antisera from three commercial PCV2a-based vaccines (vaccines A, B, and C) were used to investigate the neutralization ability of PCV2 field isolates, including one PCV2a, one PCV2b, and three PCV2d strains. Cross-neutralization was performed by first determining the neutralizing antibody titers using VNT90 with the antisera of the three PCV2a-based vaccines (A, B, and C) against QIA215, which represents the homologous PCV2a genotype. The VNT90 titers were 256, 256, and 512 for vaccine A (*n* = 3); 128, 128, and 256 for vaccine B (*n* = 3); and 128, 512, and 1024 for vaccine C (*n* = 3). Subsequently, PCV2a (QIA215), PCV2b (QIA418), and three PCV2d isolates (QIA244, QIA126, and QIA169) were neutralized using the VNT90 titer of each vaccine (Figure 3). The results showed that the neutralization efficiencies against QIA215 and QIA418 were above 95% for vaccines A, B, and C. However, the neutralization efficacies varied among the three PCV2d strains. The three PCV2a-based vaccines neutralized up to 33.7% of the QIA244 of PCV2d-1 and up to 60% of the QIA126 of PCV2d-2. In contrast, QIA169 from the PCV2d-3 subgroup showed a viral neutralization (VN) of 93.4 to 95.7%.

#### 3.1.3. Cross-Neutralization of PCV2d-VLPs against PCV2 Strains

To evaluate the cross-neutralization with antisera of PCV2d-VLPs derived from the three subgroups, VLPs of QIA244 (rQIA244), QIA126 (rQIA126), and QIA169 (rQIA169) were produced using the BES (Figure 4a) and used to immunize guinea pigs. First, NA titers were determined by VNT90 using antisera from guinea pigs immunized with rQIA244, rQIA126, or rQIA169 against the corresponding homologous virus. The VNT90 titers were 1024, 2048, and 4098 for anti-rQIA244 sera (*n* = 3); 2048, 2048, and 8192 for anti-rQIA126 sera (*n* = 3); and 1024, 8192, and 8192 for anti-rQIA169 sera (*n* = 3). Subsequently, the five PCV2 strains were neutralized against their respective VNT90 titers. The results showed that all PCV2 field strains were effectively neutralized by 90% using anti-rQIA244 serum (Figure 4b). In the case of anti-rQIA126 sera, four strains, excluding QIA244 (average, 73.8%), showed a 90% neutralization (Figure 4c). Anti-rQIA169 sera showed a 99% neutralization of QIA215 and QIA418, but only a 43.9% and a 47% neutralization of QIA244 and QIA126, respectively (Figure 4d).

### 3.2. In Vivo Protection of 2dVac and 2aVac

Based on the in vitro neutralization data of three commercial vaccines (vaccine A, B, and C) and PCV2d-VLPs, rQIA244 was selected as an experimental vaccine. The in vivo protective efficacy of rQIA244 (2dVac) against the QIA244 strain, which belongs to the PCV2d-1 subgroup, was evaluated (Figure 1). The protective efficacy of the 2dVAC was compared with that of a commercially available PCV2a-based vaccine (2aVac).

#### 3.2.1. Evaluation of Vaccine Efficacy against PCV2d Challenge

Studies on pigs were conducted to evaluate the protective efficacy of 2dVac and 2aVac against double challenge with PCV2d and PRRSV. No clinical signs, including rectal fever, diarrhea, depression, or rough hair coat, were observed in the 2dVac, 2aVac, and noVac groups 21 dpc. The body weights of the 2aVac and noVac groups were slightly lower than those of the 2dVac group, from 7 to 21 dpc (Appendix A). Similarly, the ADWG of the 2aVac (0.44 ± 0.16 kg/day) and noVac (0.50 ± 0.21 kg/day) groups was slightly lower than that of the 2dVac group (0.67 ± 0.13 kg/day) (Figure 5a). However, there were no significant differences between the groups.

Viremia was not detected or was minimal in the 2dVac and 2aVac pigs after the challenge, whereas viremia increased significantly (*p* < 0.05) in the noVac group at 14 dpc (34.3 ± 44.2 copies) and 21 dpc (244 ± 212 copies) (Figure 5b). At 21 dpc, PCV2 genomes were detected in the lung (0.07 ± 0.04 copies), tonsil (0.03 ± 0.01 copies), mesenteric LN (7.3 ± 3.8 copies), and inguinal LN (0.02 ± 0.01 copies) of the 2dVac group. A slightly higher number of PCV2 genomes was detected in the 2aVac group (3.3 ± 1.3 in the lung, 5.8 ± 1.8 in the tonsil, 1017 ± 377 in mesenteric LN, 7.0 ± 2.9 in inguinal LN) but without statistical significance. The noVac group showed significantly higher levels of PCV2 genomes than did the other groups (73 ± 33 in the lung, 272 ± 130 in the tonsil, 18,815 ± 7048 in mesenteric LN, 165 ± 64 in inguinal LN) (Figure 5c–f).

The number of macroscopic lesions of the lung and inguinal LN of the 2dVac (0.0 ± 0.0, 1.0 ± 0.0) and 2aVac (1.0 ± 0.7, 1.0 ± 0.0) groups was significantly lower (*p* < 0.05) than that of the noVac group (2.0 ± 0.5, 1.7 ± 0.5) (Appendix A). In addition, lung lesion scores were significantly different (*p* < 0.05) between the 2dVac and 2aVac groups, with interstitial pneumonia being the main observation (Appendix A). The microscopic lesions of the lung, tonsil, and mesenteric LN of the 2dVac group were not scored and were only slightly observed in inguinal LN (0.3 ± 0.2) (Figure 5g and Appendix A). The lesion scores of the lungs were significantly (*p < 0.05*) different between the 2dVac (0.0 ± 0.0) and 2aVac (1.0 ± 0.2) groups; however, the scores of other tissues did not differ significantly between these two groups. Meanwhile, the scores of the 2dVac and 2aVac groups were significantly (*p* < 0.05) lower than those of the noVac group (2.0 ± 0.3, 1.8 ± 0.2, 1.8 ± 0.2, and 1.8 ± 0.2) in all four tissues. In the histopathological analysis, mild or moderate peribronchial lymphocyte regeneration in the lungs (Figure 5h), lymphocyte loss, and macrophage proliferation in the tonsils and two LNs were mainly observed in the 2aVac and noVac groups (Appendix A).

#### 3.2.2. NA Titers of 2dVac and 2aVac against PCV2 Strains

To evaluate the cross-neutralization ability, the antisera of 2dVac, 2aVac, and noVac groups were tested against four PCV2 strains representing the PCV2a, 2b, and 2d genotypes (Figure 6). The NA titers of the 2dVac against QIA215, QIA418, QIA169, and QIA244 (4779 ± 1672, 3755 ± 836, 2219 ± 1007, and 661 ± 698, respectively) were significantly (*p* < 0.05) higher than those of 2aVac (512 ± 314, 614 ± 388, 819 ± 280, and 16 ± 9.8, respectively) and noVac (768 ± 755, 1088 ± 1052, 85 ± 33, and 8.0 ± 0.0, respectively). However, there were no significant differences in the NA titers between 2aVac and noVac against any PCV2 strain.

## 4. Discussion

Recently, there has been a genotype shift from PCV2a/2b to PCV2d, not only in South Korea but also worldwide [8,9,10,11,13,22]. Despite the continuous use of the PCV2 vaccine in Korea, PCV2 remains prevalent, with most of the detected genotypes being PCV2d. Therefore, the protective efficacy of PCV2a-based vaccines against PCV2d field strains must be evaluated.

Cases of suspected PCV2a vaccine failure have been reported [23]. It has been speculated that PCV2d may escape the protection provided by PCV2a vaccines [8,13,19]. In our previous study, we confirmed the antigenic differences between PCV2a and PCV2d by cross-neutralization experiments using PCV2 genotypes and antisera from PCV2a vaccines, as well as antigen–antibody reactivity using PCV2d isolates and PCV2a monoclonal antibodies [12]. Figure 2b shows approximately 10% differences between PCV2a and PV2d in the *orf2* nucleotide and the amino acid sequence encoding the cap protein, which is known to be antigenic. Most of these mutations were located in the epitope region described in previous studies [24,25,26]. Furthermore, neutralization experiments were performed on PCV2a, PCV2b, and PCV2d isolates using antisera from three commercial PCV2a-based vaccines, as shown in Figure 3. However, the three PCV2d isolates exhibited different neutralization efficiencies. This finding was consistent with the results of the national PCV2d genotyping analysis conducted between 2016 and 2020 (Figure 2c). Specifically, the detection rate was high in the group containing QIA244 and QIA126, which showed low neutralization efficiencies against the PCV2a vaccine sera, and the detection rate was low in the group containing QIA169. Therefore, the PCV2d field strains with a low protective efficacy against current commercial PCV2a-based vaccines are prevalent.

PCV2d is a variant of PCV2b with an additional lysine (K) added to the last codon [27]. However, PCV2d differed from PCV2b by approximately 7% in both the nucleotide and amino acid sequences. The amino acid homology among the currently prevalent PCV2d isolates in Korea was approximately 98%, with differences of one or two amino acids among the 234 amino acids of the ORF2 protein, as shown in Figure 2b. Despite these slight differences in amino acid sequences, the antisera of the PCV2a vaccines showed different degrees of neutralization efficacy against the QIA244 (0–33.7%), QIA126 (0–60.0%), and QIA169 (93.4–95.6%) isolates. The mutations in ORF2 among PCV2d isolates mainly involve differences between isoleucine (I) or phenylalanine (F) at position 53 and arginine (R) or glycine (G) at position 169. In particular, QIA169, which showed high neutralization efficacy with the PCV2a vaccine, had F at position 53 and differed from G and S at position 169. Conversely, QIA244 and QIA126, which showed low neutralization efficiencies, contained I at position 53, R in QIA244, and G in QIA126 at position 169. Overall, the difference in neutralization efficiency between the PCV2a vaccine and the PCV2d isolates was likely due to the amino acid at position 53. Previous reports have highlighted the antigenic differences resulting from single amino acid sequence variations [28], and two amino acid sites (53 and 169) located within the putative epitopes [29]. The difference in antigen–antibody reactivity may be due to a change in the three-dimensional conformation of the cap protein resulting from a single amino acid mutation. Therefore, further investigation is required to understand the effect of sequence variations at positions 53 and 169 on conformational changes in the cap protein and the resulting differences in antigenicity.

Although there is high genetic identity within the PCV2d genotype in Korea, antigenic differences between isolates have been observed in neutralization experiments. PCV2d isolates were grouped into PCV2d-1, PCV2d-2, and PCV2d-3 based on the sequence differences of one or two amino acid sites (53, 169) in the PCV3d ORF2 region associated with antigenicity. Approximately 70% of the isolates belonged to PCV2d-1 and PCV2d-2 groups (Figure 2a). PCV2d-3 showed a low detection rate from 2017 to 2018 and has not been detected since then. A cross-neutralization test with PCV2d isolates showed that antiserum from guinea pigs immunized with QIA244, a representative strain of the PCV2d-1 group, provided broad protective immunity against PCV2a, PCV2b, and other PCV2d isolates. However, QIA126, representing PCV2d-2, showed relatively low protective ability against QIA244, and QIA169 showed low protective immunity against the other PCV2d isolates. However, all three isolates were highly protective against PCV2a and PCV2b. Based on these findings, we determined that QIA244, which showed effective protective immunity against all PCV2a, PCV2b, and PCV2d genotypes, was the most suitable vaccine.

In our previous study, we reported the production of a recombinant protein vaccine using QIA244 ORF2 and its protective efficacy in piglets [20]. In this study, we evaluated the protective efficacy of PCV2d and PCV2a vaccines against PCV2d isolates. As shown in Figure 5, both 2dVac and 2aVac provided effective protection against the challenge with PCV2d (QIA244 isolate). In the 2dVac group, no virus was detected in the lungs, tonsils, or inguinal LNs, and only a small amount of the PCV2d antigen was detected in the mesenteric LNs. No significant lesions were observed in the lungs. Compared to the noVac group, the 2aVac group also showed protection against PCV2d; however, the PCV2d antigen was detected in the tissues, and significant lesions were observed in the lungs. In the cross-neutralization experiment using antiserum collected 3 weeks after vaccination, the antiserum from the PCV2d vaccine showed higher NA titers against PCV2a, PCV2b, and PCV2d than did the antiserum from the PCV2a vaccine. This may be due to the different compositions of PCV2d and PCV2a vaccines such as adjuvants and antigen doses (200 µg vs. 20 µg, respectively). Regardless of the homologous genotype, QIA169 and QIA244 of PCV2d isolates showed lower NA titers than QIA215 (2.2-fold and 6.8-fold, respectively) and QIA418 (1.7-fold and 5.7-fold, respectively) in the antiserum of 2dVac. This may be derived from antigenic difference between genotypes [12]. The monoclonal antibody used in this study was produced through the immunization of PCV2a. Therefore, the antigen–antibody reactivity can be relatively low against PCV2d isolates. The NA titers of the PCV2a vaccine were approximately 8-fold lower against QIA215, QIA418 and QIA169, and approximately 32-fold lower against the challenge strain QIA244 (PCV2d) than those of the PCV2d vaccine. The low NA titer of the antiserum derived from the PCV2a-based vaccine against QIA244 contributed to the pathogenicity of PCV2d strains in pigs and their continued circulation in pig farms.

## 5. Conclusions

The 2dVac, provided by the recombinant ORF2 protein of the QIA244 strain, which was selected as a vaccine antigen owing to its low viral neutralizing efficacy against current commercial vaccines but high cross-neutralizing ability against PCV2a, 2b and 2d strains, conferred broad protection against PCV2 genotypes in vitro and in vivo.

## Figures and Tables

**Figure 1 vaccines-11-01447-f001:**
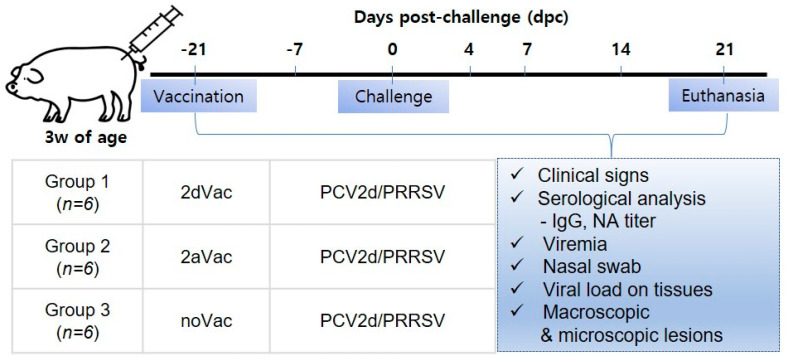
Experimental design. Eighteen piglets were randomly assigned into three groups: 2dVac (*n* = 6), 2aVac (*n* = 6), and noVac (*n* = 6). The piglets were vaccinated at −21 dpc, challenged at 0 dpc, and euthanized at 21 dpc. All piglets were dually challenged with PCV2 and PRRSV. During the whole period of experiments, sample collection and data analysis were conducted in accordance with the standard operating procedures (SOP) of the laboratory.

**Figure 2 vaccines-11-01447-f002:**
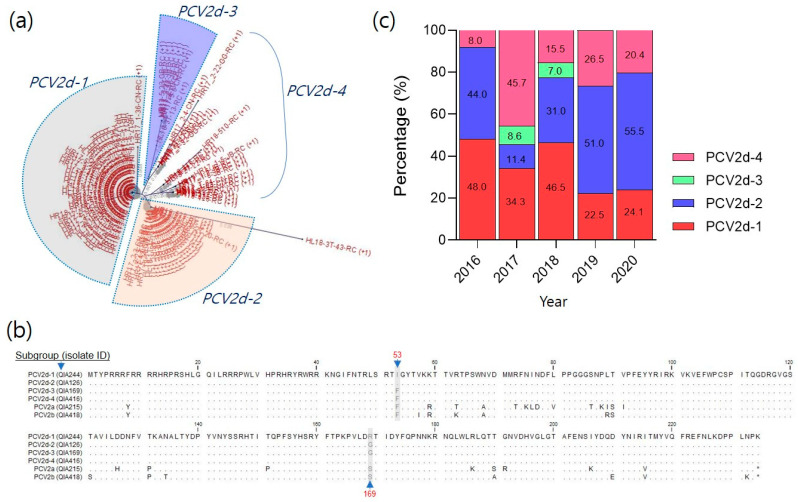
Antigenic analysis of current PCV2d field isolates in South Korea. A total of 234 PCV2d ORF2 sequences were analyzed phylogenetically using reference strains. The PCV2d isolates were classified into four subgroups based on the difference in ORF2 amino acid sequence (**a**). The representative strains from each subgroup were named QIA244 (PCV2d-1), QIA126 (PCV2d-2), QIA169 (PCV2d-3), and QIA416 (PCV2d-4). In the ORF2 amino acid sequence alignment, PCV2d isolates were divided by two unique mutation sites at positions 53 (F or I) and 169 (R or G) (**b**). Asterisks (*) in the last residue of PCV2a and PCV2b indicate a stop codon. With 234 PCV2d field isolates, the detection ratio of PCV2d subgroups was investigated from 2016 to 2020 (**c**). More detailed data are available in Appendix A.

**Figure 3 vaccines-11-01447-f003:**
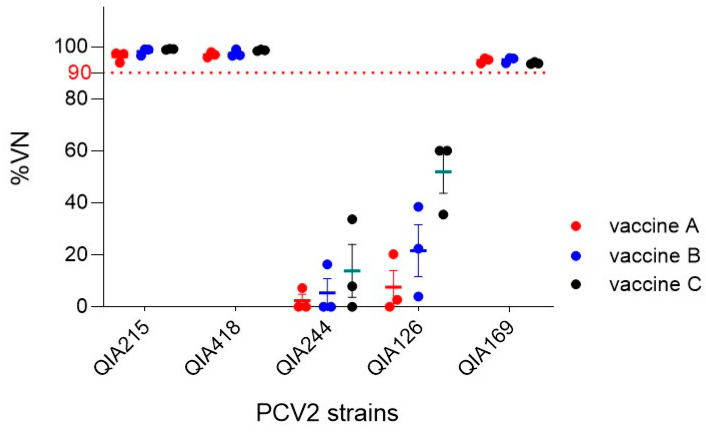
VN of three PCV2a-based vaccines. PCV2a (QIA215), PCV2b (QIA418), PCV2d-1 (QIA244), PCV2d-2 (QIA126), and PCV2d-3 (QIA169) were tested with antisera collected from pigs (*n* = 3 per vaccine). The pigs were vaccinated with PCV2a-based commercial vaccines (vaccines A, B, and C). Cross-neutralization is expressed as %VN (mean ± SE).

**Figure 4 vaccines-11-01447-f004:**
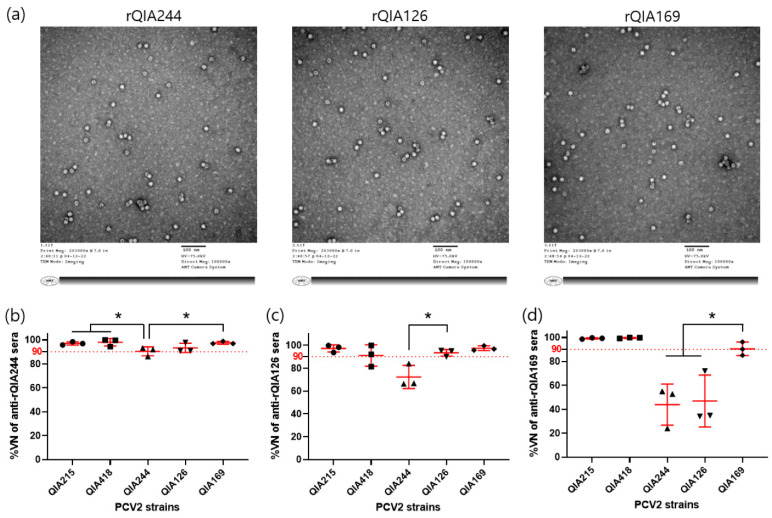
Production of PCV2d VLPs and evaluation of the cross-neutralization with antisera of PCV2d strains in guinea pigs. rQIA244, rQIA126, and rQIA169 were produced by the baculoviral expression system (**a**). PCV2a (QIA215), PCV2b (QIA418), and PCV2d (QIA244, QIA126, QIA169, and QIA416) strains were neutralized by antisera immunized with rQIA244 (**b**), rQIA126 (**c**), and rQIA169 (**d**). NA titers of anti-rQIA244, anti-rQIA126, and anti-rQIA169 sera were determined by VNT90 against the corresponding strain and applied to other strains. Cross-neutralization is expressed as %VN (mean ± SE). * indicates a significant difference (*p* < 0.05).

**Figure 5 vaccines-11-01447-f005:**
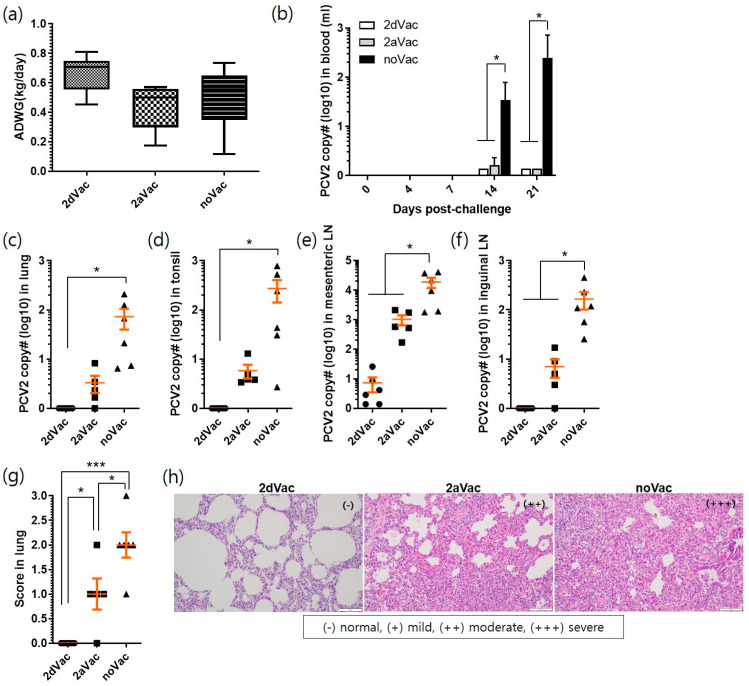
Evaluation of in vivo protection with experimental 2dVac and 2aVac. Average daily weight gain (ADWG, kg/day) was estimated for 21 days after challenge (**a**). The copy number of PCV2 was quantified in the blood of 2dVac, 2aVac, and noVac groups (**b**). The viral load of PCV2 was quantified in the lung (**c**), tonsil (**d**), mesenteric LN (**e**), and inguinal LN (**f**). The histopathological lesions were scored in the lung (**g**). The tissue sections of 2dVac, 2aVac, and noVac groups were stained using H&E (**h**), and the lesions were scored as normal (-), mild (+), moderate (++), and severe (+++). All data are expressed as mean ± SE. *, *** indicate significant differences (*p* < 0.05 and *p* < 0.001, respectively).

**Figure 6 vaccines-11-01447-f006:**
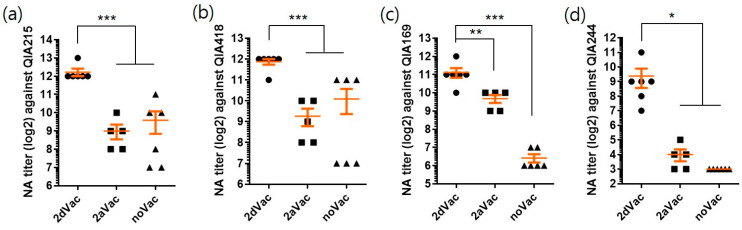
Virus neutralization assay with antisera of 2dVac, 2aVac, and noVac at 0 dpc. The NA titers were measured with antisera of 2dVac, 2aVac, and noVac against QIA215 as PCV2a (**a**), QIA418 as PCV2b (**b**), and QIA169 (**c**) and QIA244 as PCV2d genotype (**d**). NA titers are expressed as mean ± SE. *, **, *** indicate significant differences (*p* < 0.05, *p* < 0.01, and *p* < 0.001, respectively).

## Data Availability

The datasets generated or analyzed during this study are available from the corresponding author on reasonable request.

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
