# Peer review of "Selection and Evaluation of Porcine circovirus (PCV) 2d Vaccine Strains to Protect against Currently Prevalent PCV2"

_vaccines, 2023, doi:10.3390/vaccines11091447_

Round 1

Reviewer 1 Report

Dear editor of Vaccines

I hope all of you are always fine. Regarding the revision of the manuscript No. vaccines-2568737, titled “Selection and evaluation of Porcine circovirus (PCV) 2d vaccine strains to protect against currently prevalent PCV2 ”. Really, it is an interesting research; however, some comments should be replied.

Comments:

1-    Why NA titers of the 2dVac were significantly higher than 2aVac against QIA215 although it is a PCV2a strain?????? And significantly lower for QIA169, and QIA244 although are PCV2d isolates????? Please add your answer to discussion.

2-    What kind of samples were collected for Quantification of PCV2 and PRRSV? How many samples were collected and at what ages post challenge?

3-    Line 278: How could you measure Viremia??

4-    Lines 197-198: Recently, PCV2d has become increasingly prevalent in South Korea from 2016 to 2020, 234 PCV2d field strains??? Are these results of an epidemiological study you applied??? More details are needed in methodology?

5-    Two pig groups that experimentally infected with PCV2d or PRRSV-1 alone were essential. 

6-    Line 223: What about the antisera from three commercial PCV2a-based vaccines (vaccines A, B, and C)?? Were these a specific antisera? More details are needed to be mentioned in lines 164-172.

7-    Delete lines 175-176 as they previously mentioned in lines 129-130.

8-    Lines 273-277: Were these results significant or not?

Author Response

Thank you very much for giving us a chance to revise our manuscript entitled ‘Selection and evaluation of porcine circovirus (PCV) 2d vaccine strains to protect against currently prevalent PCV2’ by Ju et al. All comments made by the reviewers were quite valid and we sincerely appreciate for reviewers’ endeavor to improve the quality of our manuscript. We considered all suggestions of the reviewers for revising the original version of manuscript. We prepared a point-by-point report to explain our revision in details and also annexed a highlighted copy of the revised manuscript. However, we are still afraid of remaining of any mistake, and misunderstanding and neglecting of reviewers’ comments. We of course are ready for further revising our manuscript as the reviewers’ request. 

Thank you.
Sincerely yours,

Seok-Jin Kang

Reviewer 2 Report

The authors present an interesting study that has investigated the genetic changes of porcine circovirus 2 (PCV2) that may underpin apparent vaccine failures. Characterisation of circulating PCV2 strains over a five year period suggests that the dominant genotypes in South Korea of PCV2d is mismatched with the dominant PCV2a vaccines. The study demonstrates differences in the neutralising capacity of sera immunised with current vaccines. Using prototype virus like particle vaccines based on PCV2d, the authors were able to demonstrate enhanced protection from disease in a dual virus challenge model.

Overall, the study is very interesting. The conclusions drawn are well supported by the presented data. The manuscript is well written, though I have made some suggestions below for the authors to consider.

Line 35 – I would suggest the text “Porcine circoviruses” not be italicised. Only recognised species names should be in italics.

Line 40 – Are the authors sure that “shift” is the correct term to use here?

While this term generally used in the contexts of antigenic shift and antigenic drift, shift and drift typically refer to a rapid/dramatic and gradual/subtle changes respectively. The best example of course is influenza virus where exchange of genomic segments can result in major changes in the genotype and antigenic phenotype. For viruses with non-segmented genomes, I would think that “shift” would need to be associated with a mechanism such as recombination.

Whereas the divergence of PCV2 into genotypes appears to be a more subtle process involving amino acid substitutions, which more suggestive of “drift” in my opinion. Is there any

Line 120 – How was “low PCV2-specific IgG antibody” titre defined? I would suggest the authors add the specific cut-off used. I appreciate that given the ubiquitous nature of PCV2 it would be difficult to source virus free pigs in an economically viable manner. However, regardless of how low the cut-off titre used was, it may still have influenced the outcome of the reported study and is therefore of importance with respect to repeatability. Similarly, were these titers considered in allocating the animals to treatment groups?

Line 128 What is meant by “intranasally injected”?

Usually, intranasal installation would be achieved either by aerosol or dropwise means.

Line 130 Why were the pigs electrocuted, post-injection with pentobarbital? Was a sublethal dose of the drug used?

Line 158 The probe primer sequence would normally include a fluorophore and quencher.

Line 201 suggest replacing “orf2” with “ORF2”, this the usual naming convention for gene and polypeptide pairs. With the lowercase gene in italics.

Line 202-204 I would suggest authors round these similarity estimates to one decimal place. Given the number of amino acids in the alignment, I am not sure two decimal places are required.

Line 213 Figure 2a I am not sure of the value of this figure given that few of the details are visible. I would suggest an abbreviated version of the diagram be developed for inclusion in the main text, with the full diagram being provided as a supplemental file.

Line 213 Figure 2b In the last two positions in the alignment. What do the asterisks represent?

These typically represent stop codons. They should not be included in the alignment. Similarly, for PCV2b there is a “-“ which presumably this means a gap used to facilitate alignment. For PCV2a there is a symbol I cannot quite make out. In any case the alignment should end with the final amino acid for each of the polypeptides. Can the authors also confirm that the similarly estimates are based on amino acid residues only. If not, they should be revised to do so.

Line 238 Figure 3 – The legend should be modified to describe what the horizontal lines and error bars represent.

Line 256 Figure 4 – The legend should be modified to describe what the horizontal lines and error bars represent.

Line 305 Figure 5 – The legend should be modified to describe what the horizontal lines and error bars represent.

Line 322 Figure 6 – The legend should be modified to describe what the horizontal lines and error bars represent.

Line 327 This statement on this genotype shift should be supported by a relevant reference(s).

Line 372 suggest replacing “homology” with “identity”

The quality of the English is fine. Some suggestions in my comments to the authors.

Author Response

(The authors gave the same response as above.)
